# The Demand–Control Model as a Predictor of Depressive Symptoms—Interaction and Differential Subscale Effects: Prospective Analyses of 2212 German Employees

**DOI:** 10.3390/ijerph18168328

**Published:** 2021-08-06

**Authors:** Hermann Burr, Grit Müller, Uwe Rose, Maren Formazin, Thomas Clausen, Anika Schulz, Hanne Berthelsen, Guy Potter, Angelo d’Errico, Anne Pohrt

**Affiliations:** 1Unit 3.2 Psychosocial Factors and Mental Health, Federal Institute for Occupational Safety and Health (BAuA), 13017 Berlin, Germany; rose.uwe@baua.bund.de (U.R.); schulz.anika2@baua.bund.de (A.S.); 2Federal Institute for Occupational Safety and Health (BAuA), 10317 Berlin, Germany; grit.mueller@gmail.com; 3Unit 3.0 Work and Health, Federal Institute for Occupational Safety and Health (BAuA), 10317 Berlin, Germany; formazin.maren@baua.bund.de; 4National Research Centre for the Working Environment, 2100 Copenhagen, Denmark; tcl@nfa.dk; 5Centre for Work Life and Evaluation Studies (CTA) & the Faculty of Odontology, Malmö University, 211 19 Malmö, Sweden; hanne.berthelsen@mau.se; 6Department of Psychiatry and Behavioral Sciences, Duke University, Durham, NC 27701, USA; guy.potter@duke.edu; 7Department of Epidemiology, Local Health Unit ASL TO 3, Piedmont Region, 10095 Turin, Italy; angelo.derrico@epi.piemonte.it; 8Department of Medical Psychology, Charité-Universitätsmedizin, 10317 Berlin, Germany; anne.pohrt@charite.de

**Keywords:** interaction, superadditivity, differential effects, demand control model, skill discretion, decision authority

## Abstract

Testing assumptions of the widely used demand–control (DC) model in occupational psychosocial epidemiology, we investigated (a) interaction, i.e., whether the combined effect of low job control and high psychological demands on depressive symptoms was stronger than the sum of their single effects (i.e., superadditivity) and (b) whether subscales of psychological demands and job control had similar associations with depressive symptoms. Logistic longitudinal regression analyses of the 5-year cohort of the German Study of Mental Health at Work (S-MGA) 2011/12–2017 of 2212 employees were conducted. The observed combined effect of low job control and high psychological demands on depressive symptoms did not indicate interaction (RERI = −0.26, 95% CI = −0.91; 0.40). When dichotomizing subscales at the median, differential effects of subscales were not found. When dividing subscales into categories based on value ranges, differential effects for job control subscales (namely, decision authority and skill discretion) were found (*p* = 0.04). This study does not support all assumptions of the DC model: (1) it corroborates previous studies not finding an interaction of psychological demands and job control; and (2) signs of differential subscale effects were found regarding job control. Too few prospective studies have been carried out regarding differential subscale effects.

## 1. Introduction

A widely used psychosocial construct in occupational social epidemiology [1], the demand–control (DC) model, assumes that occupational health risks are dependent on psychological demands and job control [2,3]. The DC model—being introduced by Robert Karasek in 1979—assumes that high psychological demands, e.g., high work load, and low job control, e.g., low autonomy, in a work context pose a special health risk for workers [2,3]. It supposes that exposure to high demands without opportunity to control the work situation would cause a chronic stress reaction, which again would lead to non-somatic and somatic disease outcomes [2,3].

The DC model combines psychological demands and job control into four categories, or quadrants: (i) the strain quadrant has high demands and low job control; (ii) the low strain category has low demands and high job control; (iii) the active quadrant has high demands and high job control; and (iv) the passive quadrant has low demands and low job control. The strain quadrant is proposed to have the highest risk for poor health and the low strain quadrant is proposed to have the lowest risk, whereas the active and passive quadrants would have intermediate risks [3,4]. The DC model supposes that the observed health effect of the strain quadrant would be higher than the additive effect of low job control and high psychological demands (i.e., superadditivity), that is, the model supposes interaction [3,4].

According to the DC model, psychological demands describe the general workload in an occupational context [3]. Job control denotes to what extent autonomy can be exercised through decision authority, e.g., influence at work, and skill discretion, e.g., opportunities for development. Decision authority denotes the extent to which one can influence decisions at the workplace. Skill discretion deals with the opportunity to learn and develop skills at work. Psychological demands have mainly been measured by items on work pace and amount of work (together often called quantitative demands), whereas job control is defined by the decision authority and skill discretion (e.g., opportunities for development) [4,5]. In this paper, we label work pace and amount of work on the one hand and decision authority and skill discretion on the other hand as subscales of the main scales of psychological demands and job control, respectively. The approach of combining these subscales is justified if these subscales have the same association with outcomes as their main scales [1].

Since the introduction of the DC model [2], research on that model has considered depressive symptoms as an outcome [1,6,7]. Even if ample prospective research has established job strain as a risk factor for depressive symptoms [6,7], two issues are still unresolved: (1) we still do not know if this elevated risk is just the sum of the risk of high psychological demands and low job control or if there is a surplus risk over and above their mere sum of risks [1,6,8,9,10]; and (2) we also do not know if health effects of the subscales work pace and amount of work are similar, and if health effects of the subscales decision authority and skill discretion are similar. A systematic investigation of possible differential subscale effects of psychological demands and job control on health has to our knowledge seldom been carried out [1,11,12].

On the basis of these considerations, we tested the following assumptions of the DC model:Regarding the supposed interaction of psychological demands and job control, the combined effect of high psychological demands and low job control on depressive symptoms is stronger than their mere sum (i.e., superadditivity).Regarding differential effects on depressive symptoms of the subscales of psychological demands and job control: 2.a. the subscales of work pace and amount of work have similar associations with depressive symptoms; 2.b. the subscales of decision authority and skill discretion have similar associations with depressive symptoms.

## 2. Materials and Methods

### 2.1. Study Overview

In a cohort of employees in Germany with self-reported data from 2011/12 and 2017, we carried out binomial regressions with depressive symptoms as the dependent variable in order to test the interaction between the independent variables, psychological demands and job control, and possible differential effects of subdimensions of the independent variables.

### 2.2. Population

We used data from the German Study on Mental Health at Work (SMGA), which is a nation-wide representative employee cohort study with a baseline survey in 2011/12 and a follow-up in 2017 [13], mean follow-up time 5.1 year (minimum 4.7 years, maximum 5.4 years). At the baseline, the target population consisted of all employees subjected to compulsory social security in Germany on 31 December 2010 born in 1951–1980 [13]. The study population was enrolled through the register of Integrated Employment Biographies (IEB) of the German Federal Employment Agency at the Institute for Employment Research (IAB). This register does not cover civil servants, self-employed workers and freelancers. A comparison of the analyzed cohort with the sample frame shows that people with a lower age and lower social class had a slightly lower response, whereas people with a higher age and higher social class had a slightly higher response; there were no differences due to gender [13,14]. The analyzed cohort comprised 2212 people being employees at the baseline (Figure 1).

In the cohort analyzed, women constituted half of the sample, most participants were skilled workers (International Standard Classification of Occupations, ISCO, main groups 4–7, regarding ISCO see last paragraph of the independent variable Section 2.3.2), and mean age was 47 years (Table 1). Average age was 46.7 years. The median score for psychological demands was 2.3—for the subscales work pace and amount of work, the scores were 3.0 and 1.9, respectively—and the job control scale median was 2.3, for the subscales of decision authority and skill discretion, the scores were 1.8 and 2.8, respectively (Table 1). Employees were distributed evenly in the four quadrants of strain, no strain, passive, and active, reflecting a very low association between psychological demands and job control (Pearson correlation −0.033).

### 2.3. Variables

Information at the baseline on gender, age, socioeconomic position, psychological demands, and job control was collected via a computer-assisted personal interview. At the end of the baseline and follow-up interviews, participants answered items on depressive symptoms privately in a separate paper questionnaire, which they gave back to the interviewer in a closed envelope to ensure confidentiality. Regarding the scales described below, quartiles, Cronbach’s alphas, and inter-item correlations are shown in Table 1.

#### 2.3.1. Dependent Variable

Depressive symptoms were assessed using the German version of the 9-item scale of the Patient Health Questionnaire (PHQ) [15,16]. Symptoms were measured with the question: “Over the last 2 weeks, how often have you been bothered by any of the following problems?” The nine items in the scale were: “Little interest or pleasure in doing things”, “Feeling down, depressed or hopeless”, “Difficulty falling asleep or sleeping or increased sleep”, “Tiredness or feeling unable to have energy”, “Decreased appetite or excessive need to eat”, “Bad opinion of yourself”, “Difficulty concentrating on something”, “Slowed speech/movement or restlessness (“fidgety”)”, and “Thoughts that you would rather be dead or want to self-inflict pain”. Response options to these questions were: “Not at all” (0), “Several days” (1), “More than half the days” (2), and “Nearly every day” (3). The scale score for depressive symptoms was computed as the sum of all items [15]. If not all, at least half of the items were answered and an adjusted sum was calculated, taking the number of missing items into account. The scale thus ranged from 0 to 27. The Cronbach’s alpha of the published German version was 0.88, which is on the same level as in the present study (Table 1) [15]. For our analyses, we treated depressive symptoms as a dichotomous variable with a cut-off of 10 or above as the screening threshold for major depressive disorder [17]. Of the participants, 8% experienced depressive symptoms at the baseline and 10% at follow-up (PHQ-9 ≥ 10).

#### 2.3.2. Independent Variables

Psychosocial working conditions were measured with items from the 1st version of the Copenhagen Psychosocial Questionnaire (COPSOQ 1) [18,19]. The COPSOQ instrument enables the measurement of various constructs, including the DC model. A comparison of the original items from the Job Content Questionnaire (JCQ) developed to measure the scales from the DC model and the corresponding items from the COPSOQ instrument used in the present paper can be seen in Table 2.

The COPSOQ items had the following response options: “Always” (4), “Often” (3), “Sometimes” (2), “Seldom” (1), and “Never/hardly ever” (0), apart from two items in the scale regarding skill discretion (see below). For the main analysis of interaction, all scales, ranging from 0 to 4, were dichotomized at their medians (median values can be seen in Table 1). For a sensitivity analysis, the main scales of psychological demands and job control were dichotomized at the value 2; high demands were ≥2–4, low control was 0–≤2. This value was chosen as it was in the middle of the scale range from 0 to 4. For the main analyses of subscale effects, subscales (work pace, amount of work, decision authority, and skill discretion) were also dichotomized at their medians (median values can be seen in Table 1) [2] and divided into three categories based on the value ranges 0–<1.333 for “Low”, 1.333–<2.666 for “Medium” and 2.333–4 for “High”—for a sensitivity analysis, see the analysis subsection below. These ranges were chosen so that each covered one third of the value ranges, which, as indicated above, range from 0 to 4.

Psychological demands: The main scale of psychological demands was based on the overall mean of the subscales work pace and amount of work described below (Table 2), such as the Job Content Questionnaire instrument (JCQ) measuring the DC model [4,5]. Cronbach’s alpha for this scale was on the same level as for a scale with the same items in a German validation study amounting to 0.82 (Table 1) [18]. The subscale of work pace was measured with one item (Table 2). The subscale of amount of work was based on the mean of four items (Table 2).

Job control: The main scale of job control (namely, decision latitude) was calculated as the mean of the subscales of decision authority and skill discretion, as described below, in order to accommodate the JCQ instrument [4]. Items of the two subscales below were given different weights in order to correspond with the JCQ measure (Table 2). The subscale decision authority was based on the mean of four items (Table 2). Cronbach’s alpha for this subscale was somewhat higher than a scale with the same items in a German validation study amounting to 0.64 (Table 1) [18,19]. The subscale of skill discretion was based on the mean of three COPSOQ items (Table 2). Two items (“learning new things” and “use skills or expertise”) had the response options and values for the scale: “To a very large extent” (4), “To a large extent” (3), “Somewhat” (2), “To a small extent” (1), “To a very small extent” (0). Cronbach’s alpha for the skill discretion subscale was on the same level as a similar scale of a German validation study, being 0.70 (Table 1) [18,19].

DC quadrants: A categorical variable was created in order to measure exposure to combinations of psychological demands and job control with the following categories defined by median values in the population [2]: “Strain” (i.e., exposed to both high psychological demands and low job control: ≥median psychological demands and <median job control); “Active” (i.e., exposed to high psychological demands and high job control: ≥median psychological demands and ≥median job control); “Passive” (i.e., exposed to low job control and low psychological demands: <median psychological demands and <median job control); and “Low strain” (i.e., not exposed; <median psychological demands and ≥median job control), the latter being the reference group. For a sensitivity analysis, a variable was created where the quadrants were defined by value ranges of the scales, where high demands corresponded to a score of 2–4, low demands 0–≤2, low job control 0–2 and high control >2–4.

Covariates: We included gender, age, and socioeconomic status. Age was treated with a linear and a quadratic term, as it has been shown to have a reversed u-shaped association with depressive symptoms in adult working populations, with the lowest levels for the youngest [20,21]. Socioeconomic status was treated as a categorical variable [22]. Occupations were manually coded according to the International Standard Classification of Occupations (ISCO 08) and categorized into four groups on the basis of skill level, following the International Standard Classification of Education (ISCED) [23]. There were two exceptions: managers (ISCO main group 1) were put together with professionals (main group 2), and plant and machine operators (main group 8) were put together with elementary occupations (main group 9).

### 2.4. Analyses

We carried out two types of analyses in order to test assumptions of the DC model:

We performed interaction analyses in order to test if the observed combined effect of high psychological demands and low job control on depressive symptoms was stronger than their expected sum (i.e., superadditivity).

We performed differential subscale effect analyses in order to see if there were effects on depressive symptoms from the subscales of psychological demands (i.e., work pace and amount of work) and job control (i.e., decision authority and skill discretion) were the same.

#### 2.4.1. Interaction Analyses

Prospective multiple binomial regression analyses were conducted, as this approach yields Rate Ratios (RR) needed for assessing interaction [24]. In the main analysis, we tested for an interaction of baseline psychological demands and job control with depressive symptoms at follow-up as the dependent variable, adjusted for the following baseline variables: depressive symptoms, socioeconomic position (treated categorically also in all other regressions described below), gender, and age. We used Rothman’s Relative Excess Risk due to Interaction (RERI) approach [25,26] to test whether strain had a higher risk than the sum of the risks of high psychological demands and low job control (i.e., superadditivity), as the DC model proposes [2,3]. REHRI was calculated according to this approach [27]. A significant positive RERI indicates superadditivity, a negative RERI, subadditivity. In a first sensitivity analysis, the main analysis was repeated, excluding those who had depressive symptoms at the baseline. In a second sensitivity analysis, the main analysis was repeated, with high psychological demands and low job control defined by value ranges (see variable description above) instead of medians, as in the main analysis.

#### 2.4.2. Differential Subscale Effect Analyses

Through a log binomial regression, we tested whether each of the baseline subscales of psychological demands (work pace and amount of work) and job control (decision authority and skill discretion)—adjusted for the same covariates as described above—had similar effects on depressive symptoms at follow-up. More specifically, we tackled the question of whether work pace and amount of work, dichotomized at their median, had similar associations with depressive symptoms and whether decision authority and skill discretion had similar associations with depressive symptoms, respectively. In a sensitivity analysis, these analyses were repeated with subscales divided into three categories based on value ranges (see variable description above), in contrast to the median criterion in the main analysis. Differences between RRs of depressive symptoms for the subscales in each dimension were tested assessing heterogeneity of the RRs through fixed-effect meta-analysis [28].

#### 2.4.3. Significance Level and Software Used

The significance level was 0.05. No Bonferroni adjustment for multiple tests was applied [29]. Data were analyzed by means of SPSS 25 (IBM SPSS, Chicago, IL, USA) using the GENLIN command in order to perform the aforementioned binomial regressions [24,25,26,27].

## 3. Results

### 3.1. Interaction Analysis

Employees in the strain quadrant had significantly increased RRs for depressive symptoms at follow-up as compared to the low strain quadrant (RR = 1.51, 95% CI = 1.03; 2.23) (Table 3). Being in the passive or active quadrants was not significantly associated with depressive symptoms. The observed combined effect of low control and high demands did not deviate from the expected effect (RERI = −0.2; 95% CI = −0.85; 0.43), that is, no superadditivity was found (Table 3, last two columns). The sensitivity analysis yielded the same results when excluding employees with baseline depressive symptoms (Appendix A, Table A1) or when dichotomizing psychological demands and job control based on scale value ranges (Appendix A, Table A2).

### 3.2. Differential Subscale Effect Analyses

#### 3.2.1. Psychological Demand Subscales

The main analysis of risk for depressive symptoms using a median criterion of categorization of the subscales work pace and amount of work found no differential subscale effects (*p*-value for heterogeneity: 0.92) (Table 4). The sensitivity analysis using a categorization based on value ranges also did not show clear differential subscale effects; although a significantly increased risk was found among those in the highest exposure group for amount of work, this risk estimate was not significantly different from that of high work pace (*p*-value for heterogeneity: 0.33) (Appendix A, Table A3).

#### 3.2.2. Job Control Subscales

The main analysis using a median criterion of categorization of the subscales decision authority and skill discretion found no differential subscale effects (*p*-value for heterogeneity: 0.58). The sensitivity analysis using a categorization based on value ranges found differential subscale effects; however, they were not strongly significant (*p*-value for heterogeneity: 0.04) (Appendix A, Table A3).

## 4. Discussion

The present study adds to a handful of studies investigating the rationale of the DC model when predicting depressive symptoms regarding (a) the interaction of the dimensions of psychological demands and job control, and (b) the differential effects of subscales of these two dimensions.

First, the present study indicates that job control does not interact with psychological demands, contrary to the expectation of the DC model: we did not observe a combined effect on depressive symptoms above the additive effect of high psychological demands and low job control, i.e., superadditivity.

Second, the present study found signs of differential effects of the subscales of job control, contrary to the DC model’s expectation of similar effects. While in the main analysis (when dichotomizing subscales at the median), differential effects of the subscales were not found (Table 4), a sensitivity analysis (when categorizing subscales based on value ranges) suggests differential effects for subscales of job control (decision authority and skill discretion). 

### 4.1. Comparison with Other Studies

Three prospective studies and one meta-analysis have investigated whether the combined effect of high psychological demand and low job control poses an additional risk over and above their main effects on depressive symptoms, i.e., whether the risk factors of job control and psychological demands interact [6,8,9,10]. None of the studies found statistical interaction. However, all four analyses did investigate deviations from multiplicativity, which is a more conservative approach than the present study’s analysis of deviations from additivity [25]. The latter approach fits better with what the JCQ model proposed, namely, superadditivity [2,3]. The three original studies were based on 2821 workers from nine companies in Belgium (Belstress), 3366 workers of a Finnish representative sample (the health 2000 study), and 11,552 workers in an energy production company in France (Gazel), respectively [8,9,10]. The meta-analysis relied on 120,211 workers from 14 cohort studies carried out in the UK, Denmark, Finland, and Sweden, none of them being among the above-mentioned original studies [6]. All three original studies used items from the JCQ instrument on psychological demands and job control [4], while the meta-analysis used a previous harmonization of measures of job strain [30]. For the present study, items from the COPSOQ similar to those from the JCQ were chosen (Table 1). To measure depressive symptoms, Belstress and Gazel used the CES-D instrument, a valid indicator of depressive symptoms, just like the present study’s PHQ-9-instrument. The health 2000 study used prescribed antidepressant medication as an indicator. The meta-analysis relied largely on data on clinical depression. In conclusion, our results regarding the absence of interaction appear to be supported by all four studies. It should, however, be noted that these studies’ approaches were more conservative, as they looked for a multiplicative interaction.

With respect to the differential subscale effects of the DC model on depressive symptoms, we are only aware of two studies that examined this question—and only regarding the job control scale [11,12]. One study was based on a random sample of the adult population in Stockholm County, Sweden, comprising 4710 workers (PART study); the other study was conducted on 10,308 civil servants in London, UK (Whitehall study). Both studies investigated the differential effects of decision authority and skill discretion. In the PART study, risks did not differ from each other as CIs were relatively wide [11]. The larger Whitehall study found borderline different subscale effects; the point estimate of low decision authority (men: RR = 1.29, 95% CI 1.1–1.5; women: 1.37, 95% CI 1.1–1.8) was in the upper end of the CIs of low skill discretion (men: 0.9–1.3; women: 0.8–1.4) [12]. Both studies used the JCQ [4]; the PART study categorized subscales based on value ranges, whereas the Whitehall study categorized them using tertiles. The PART study measured depressive symptoms by means of the Major Depression Inventory, which specifically targets depression, while the Whitehall study used the General Health Questionnaire, which measures common psychological disorders. In summary, one high powered study did not find differential subscale effects; another higher-powered study found only borderline significant effects in the same direction as the present study.

### 4.2. Strengths and Weaknesses

A strength of the present study is that it is prospective, making it easier to assess causality. Second, in order to reduce bias due to the mode of data collection, questions on depressive symptoms using the Patient Health Questionnaire were filled out by the respondent in a separate room and put into an envelope, so as to minimize an interviewer bias. Third, the study did not comprise workers older than 60 years at the baseline. As health selection occurs more often in higher age groups, this age restriction may have limited healthy worker selection already at the baseline. Fourth, we followed up all participating employees in the study disregarding if they still worked or exited work at follow-up. Depressive symptoms have been associated to exit from work [31,32]. Keeping participants who changed jobs or exited work at follow-up contributes to avoiding healthy worker selection during follow-up in the analyses of the present paper.

The strengths of this study need to be balanced against its weaknesses. First, this study is observational, and thus, as always in such studies, selection bias has to be considered. Second, the participation rate in the study is low. Based on comparisons with the study’s sampling frame, we acknowledge that there may be biases due to regional characteristics, gender, age, education, profession, and income [13,14]. Additionally, we cannot rule out biases due to possible higher attrition among those becoming depressed during follow-up. Third, although the study comprises over 2000 participants, it has limited power to detect smaller effects. A post hoc power analysis shows that the present study has a power of 0.53 to find a significant elevated risk of 1.5 [33]. However, the present study’s results could contribute to future meta-analyses. Fourth, the study did not use the original JCQ items, but used the COPSOQ tool [18,34] (Table 2). Therefore, we weighted items for the scales in the present study in order to reflect the items in the original JCQ instrument (see weights in last column of Table 1). It can be questioned that we categorized the JCQ item “My job requires working very hard” as a measure of work pace, as this item could also be considered to measure high amount of work. Using COPSOQ to measure work pace, we only had one item available, namely “Do you have to work very fast?”. However, the single work pace item has been shown to have a good correlation with the complete COPSOQ work pace scale [35].

## 5. Conclusions

The findings of the present study have implications for practice and research.

From a practical perspective, the results of the present study indicate that if one aims to prevent work-related depressive symptoms, one should reduce psychological demands and increase job control. Further, there seems to be no extra risk reduction for those exposed to both high psychological demand and low job control over and above their additive effect. Organizations should not only focus on those workers being exposed to both these risk factors, but also on those only exposed to high demands or low control. When addressing the two subdimensions of job control, namely, decision authority and skill discretion, more efforts should focus on increasing decision authority than skill discretion in order to prevent work-related depressive symptoms.

From a research perspective, it should be further investigated if the combination of high psychological demands and low job control poses an additional risk over the sum of their main effects. Apart from the present study, only four previous prospective studies have investigated this issue [6,8,9,10]. It should also be investigated if there are differential effects of the subscales of work pace and amount of work of the job demands scale and of the subscales of decision authority and skill discretion of the job control scale. The present study is the only one to have investigated the subscales of psychological demand with regard to job control; this issue has only been studied in three prospective studies (the present study and two previous studies [11,12]). Therefore, we suggest re-analyzing existing longitudinal studies, preferably in a meta-analytic fashion to overcome the limited power offered by most psychosocial cohorts, containing items covering the DC model to confirm or reject the findings of existing studies.

## Figures and Tables

**Figure 1 ijerph-18-08328-f001:**
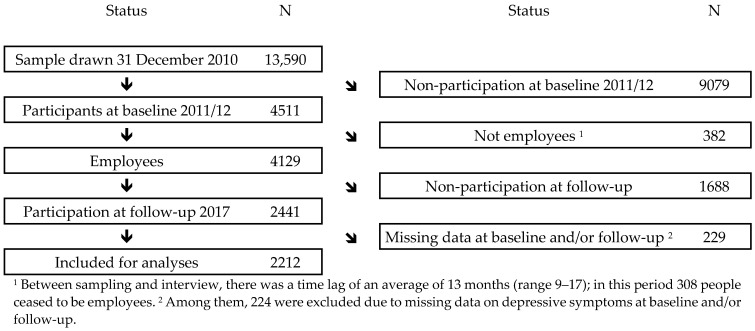
Flow diagram of participation in SMGA’s 2011/12 baseline and the 2011/12–2017 cohort.

**Table 1 ijerph-18-08328-t001:** Distribution of covariates, psychosocial working conditions at baseline and depressive symptoms at baseline and follow-up among 2212 employees aged 31–60.

Variables	N	%	1st Quartile	Median	3rd Quartile	Cronbach’s Alpha	Inter Item Correlation Range
Gender							
Men	1079	49					
Women	1133	51					
Age			41	47	53		
Socioeconomic position							
Un- and semiskilled workers, category (ISCO ^1^ main group 8,9)	255	12					
Skilled workers, category (ISCO ^1^ main group 4–7)	784	35					
Semi-professionals, category (ISCO ^1^ main group 3)	629	28					
Professionals/managers, category (ISCO ^1^ main group 1,2)	544	25					
Demand control quadrants ^2^							
Low strain, category ^2^	563	26					
Passive, category ^2^	568	26					
Active, category ^2^	532	24					
Strain, category ^2^	549	25					
Psychological demands, scale ^3^			1.8	2.3	2.8	0.82	0.26–0.67
Work pace, subscale ^3^			2.0	3.0	3.0	n/a ^4^	n/a ^4^
Amount of work, subscale ^3^			1.3	1.8	2.5	0.84	0.48–0.67
Job control, scale ^3^			1.8	2.3	2.8	0.75	0.14–0.45
Decision authority, subscale ^3^			1.1	1.8	2.4	0.71	0.32–0.43
Skill discretion, subscale ^3^			2.2	2.8	3.3	0.70	0.41–0.45
Depressive symptoms at baseline, scale ^5^			2.0	4.0	6.0	0.83	0.21–0.51
No, category (PHQ-9 < 10) ^6^	2039	92					
Yes, category (PHQ-9 ≥ 10) ^6^	173	8					
Depressive symptoms at follow-up, scale ^5^			2.0	4.0	6.0	0.83	0.18–0.55
No, category (PHQ-9 < 10) ^6^	1989	90					
Yes, category (PHQ-9 ≥ 10) ^6^	223	10					

^1^ ISCO, International Standard Classification of Occupations, see last paragraph of the Section 2.3.2 “Independent variables” [23]. ^2^ Defined by medians of psychological demands and job control (see also variable subsection of the method section). ^3^ Minimum = 0, Maximum = 4. ^4^ One item measure. ^5^ Minimum = 0, Maximum = 27. ^6^ Values for the scale.

**Table 2 ijerph-18-08328-t002:** DC model items and corresponding COPSOQ items for scales and subscales being used in the present paper.

Job Content Questionnaire (JCQ) Items, Version R-1.11 [4]			Copenhagen Psychosocial Questionnaire (COPSOQ) Items in the German Study on Mental Health at Work (S-MGA) [18]	
Scale	Subscale	Item	Item	Item Weight for Subscale and Scale
Psychological demands ^1^	Work pace ^2^	My job requires working very fast	Do you have to work very fast?	4
		My job requires working very hard		
	Amount of work ^2^	I am not asked to do an excessive amount of work ^3^	Is your workload unevenly distributed so it piles up?	1
			Do you fall behind with your work?	1
		I have enough time to get the job done ^3^	How often do you not have time to complete all your work tasks?	1
			Do you have enough time for your work tasks?	1
Job control	Decision authority	My job allows me to make a lot of decisions on my own	Do you have a large degree of influence concerning your work?	3
		I have a lot of say about what happens in my job	Do you have a say in choosing who you work with?	2
			Do you have any influence on what you do at work?	2
		In my job, I have very little freedom to decide how I do my work ^3^		
			Can you influence the amount of work assigned to you?	2
	Skill discretion	My job requires that I learn new things	Do you have the possibility of learning new things through your work?	4.5
		I have an opportunity to develop my own special abilities		
		My job requires me to be creative		
		My job requires a high level of skill	Can you use your skills or expertise in your work?	1.5
		My job involves a lot of repetitive work ^3^	Is your work varied?	3
		I get to do a variety of different things on my job		

The JCQ is the official demand–control (DC) model questionnaire [4]. ^1^ The scale also has an item on conflicting demands, which has not been considered in the present paper [5]. ^2^ Subscale identified in factor analyses [5]. ^3^ Reverse coded.

**Table 3 ijerph-18-08328-t003:** Associations between the Demand–Control (DC) measure at baseline in 2011/12 and depressive symptoms at follow-up in 2017 among 2212 employees aged 31 to 60 years in Germany. Multiple binomial regression. Rate Ratios (RR’s).

DC Quadrants (Based on Medians)	N	Observed Prevalence of Depressive Symptoms at Follow-Up (%)	*p* ^1^	RR ^2^	95% CI	RERI ^2,3^	95% CI
			0.204				
Low strain	563	6		1			
Passive	568	11		1.42	0.96; 2.10		
Active	532	9		1.31	0.87; 1.96		
Strain	549	14		1.51	1.03; 2.23	−0.21	−0.85; 0.43

^1^ This *p*-value denotes to what extent the categorical DC measure is associated with depressive symptoms at follow-up. ^2^ Adjusted for depressive symptoms, socioeconomic position, gender and age at baseline. ^3^ A positive value (i.e., a value above 0) would express superadditivity, that is, the observed combined effect of low control and high demands would be above the additive effect of these two factors [25,26]. CI = Confidence Intervals. RERI = Relative Excess Risk due to Interaction.

**Table 4 ijerph-18-08328-t004:** Associations between the DC measure subscales at baseline 2011/12 and depressive symptoms at follow-up 2017 among 2212 employees aged 31 to 60 years in Germany. Multiple binomial regression. Rate Ratios.

Dimension	Subscale	N	Observed Prevalence of Depressive Symptoms at Follow-Up (%)	*p* ^1^	RR ^2^	95% CI
Psychological demands	Work pace			0.267		
	<median ^3^	822	9		1	
	≥median ^3^	1390	11		1.15	0.90; 1.48
	Amount of work			0.201		
	<median ^3^	1202	8		1	
	≥median ^3^	1010	12		1.17	0.92; 1.49
Job control	Decision authority			0.037		
	<median ^3^	1123	13		1.30	1.01; 1.66
	≥median ^3^	1089	8		1	
	Skill discretion			0.180		
	<median ^3^	1022	12		1.18	0.93; 1.51
	≥median ^3^	1190	8		1	

^1^ This *p*-value denotes to what extent the categorical DC measure is associated with depressive symptoms at follow-up. ^2^ Adjusted for baseline depressive symptoms, socioeconomic position, gender and age. Not adjusted for the other DC measure subscales. ^3^ The median value can be seen in Table 2.

## Data Availability

A scientific use file (SUF) containing the baseline data is available at the Research Data Centre of IAB; a SUF containing the cohort is underway.

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
