# Peer review of "The Demand–Control Model as a Predictor of Depressive Symptoms—Interaction and Differential Subscale Effects: Prospective Analyses of 2212 German Employees"

_ijerph, 2021, doi:10.3390/ijerph18168328_

Round 1
Reviewer 1 Report
This paper addresses job control and depression in a large sample of German employees and was a very interesting read.
Abstract
- More background is needed in your abstract – perhaps just one or two more sentences setting the scene of what your paper is on.
- It would also be helpful to include just a little more information on your methods. For example, subscales are discussed – but the reader is not sure what subscales are being referred to.
- The key findings and implications of the study could be presented more clearly.
Introduction
- Some of the terms used in the introduction could be defined more clearly e.g., decision latitude, the demand-control model)
- The relationship between DC and depressive symptoms could be clearer. The authors seem to state that there is both a well-established link, and that no research has prospectively examined this outcome. Could this be clarified?
- The term ‘subscale’ is used consistently. Does this refer to a specific questionnaire subscale or simply a part of the DC?
Methods
- What does ISCO mean? Can the full term be used prior to the acronym
- Was the same exact survey used at each time point?
- (line 100) As in my comment above, there are scores described for ‘subscales’ – what scale does this refer to? What would be considered a ‘high’ score? Can more information be provided please.
- Line 107 – was the entire data collection based on computer assisted interview? Us tus tge sane as the SMGA?
- Table 1: This is a little confusing to understand as half only have N and %, whereas the other half have Cronbach’s etc. Can this be split into two tables? More clarity is needed. Also for demographic information (e.g., age) means and standard deviations would be helpful.
- What is the COPSOQ inventory? What does this acronym stand for?
- Table 2: this could use some more explanation. I don’t understand the difference between the JCQ and the COPSOQ and why their items are being compared (?)
- The analysis section could be explained more clearly.
Results
- The following text appears to be in error: This section may be divided by subheadings. It should provide a concise and precise 250 description of the experimental results, their interpretation, as well as the experimental 251 conclusions that can be drawn.
- Table formatting appears off in some tables
Discussion
- I suggest rearranging the order of your discussion to have the ‘comparison with other studies’ section before your limitations section.
- What if the participants were not in the same job during both data collection periods?
- Why is the study described as being underpowered? To me this seems like a large enough sample size. Have you done power analyses?
- There is not a great deal of discussion of the present findings and their implications. I suggest including this so the reader understands what to take from these findings and how they may be applied.
Author Response
July 10th, 2021
ANSWERS TO COMMENTS OF REVIEWER 1
Thank you for your valuable comments improving the quality of the manuscript. Below are our point by point answers to each of your comments (with references to relevant pages and line numbers in parentheses). In the revised version of the manuscript, changes in text are highlighted with yellow background.
- Abstract
More background is needed in your abstract – perhaps just one or two more sentences setting the scene of what your paper is on.
AUTHORS’ RESPONSE (page 1, line 24-25): We now start the first sentence indicating we test assumptions of a widely used model in psychosoial epidemiology. The first part now reads: ‘Testing assumptions the widely used demand-control (DC) model in occupational psychosocial epidemiology,( …).’ Due to the 200 word limit of the abstract, we only did that.
- It would also be helpful to include just a little more information on your methods. For example, subscales are discussed – but the reader is not sure what subscales are being referred to.
AUTHORS’ RESPONSE (page 1, line 28): We now clearly state that the regression analyses were logistic and longitudinal. Due to the 200 word limit we did not specify additional details of the analyses. We agree it would have been fine to mention the subscale names but don’t see what to delete to make space for this in the abstract. Not that all subscales are already mentioned as keywords.
- The key findings and implications of the study could be presented more clearly.
AUTHORS’ RESPONSE (page 1, line 34-38): We reworded the concluding part of the abstract in the following way: ‘This study does not support all assumptions of the DC model: 1) It corroborates previous studies not finding an interaction of psychological demands and job control. 2) Signs of differential sub-scale effects were found regarding job control. Too few prospective studies have been carried out to conclude on differential subscale effects.’
- Introduction
Some of the terms used in the introduction could be defined more clearly e.g., decision latitude, the demand-control model)
AUTHORS’ RESPONSE (page 2, lines 45-50 and 61-71): We have now reworded and extended the 1st and 3rd subsection of the introduction in order to explain central terms.
- The relationship between DC and depressive symptoms could be clearer. The authors seem to state that there is both a well-established link, and that no research has prospectively examined this outcome. Could this be clarified?
AUTHORS’ RESPONSE (page 2, lines 75-81): The existing paragraph has been reworded so as to clarify that there on the one hand is evidence for an association between job strain and depressive symptoms, but that on the other hand tests of the assumptions of the DC model are almost lacking. The paragraph now reads: “Since the very introduction of the DC model [2], research on that model has considered depressive symptoms as an outcome [1, 6, 7]. Even if ample prospective research has established job strain as a risk factor for depressive symptoms [6, 7], two issues are still unresolved: 1) We still do not know if this elevated risk is just the sum of the risk of high psychological demands and low job control or if there is a surplus risk over and above their mere sum of risks [1, 6, 8-10] . 2) We also do not know if health effects of the subscales work pace and amount of work are similar, and if health effects of the subscales decision authority and skill discretion are similar; a systematic investigation of possible differential subscale effects of psychological demands and job control on health has to our knowledge seldom been carried out [1, 11, 12].”
- The term ‘subscale’ is used consistently. Does this refer to a specific questionnaire subscale or simply a part of the DC?
AUTHORS’ RESPONSE (page 2, line 69-72): In the 3rd paragraph of the introduction we now explain what we mean by the term subscale: “In this paper, we label work pace and amount of work on the one hand and decision authority and skill discretion as subscales of the main scales psychological demands and job control respectively. The approach of combining these subscales is justified if these subscales have the same association with outcomes as their main scales have [1].”
In the variables subsection of the method section (page 4, lines 165-169), in Table 1 (Page 5) and in the conclusions section (Page 12, lines 415 -424) we are also more explicit regarding what those subscales are.
- Methods
What does ISCO mean? Can the full term be used prior to the acronym
AUTHORS’ RESPONSE: We now in the population subsection of the method section (Page 3, line 119-210) and in Table 1 (Page 5) refer to the variable subsections (2.3.2) description of ISCO. Additionally we write out the acronym and give a reference to ISCO under Table 1.
- Was the same exact survey used at each time point?
AUTHORS’ RESPONSE (Page 4, lines 128-131): With a few exceptions not relevant for the present paper, the same survey was carried out at both time points. In relation to the variables used for the present paper, depressive symptoms at baseline and follow-up were measured using the same mode of data collection and with the same items, namely by means of a paper questionnaire containing the PHQ-9 instrument items given to the respondents after the personal computer assisted interviews. Information on the independent variables were only used from the baseline personal interviews. We revised the introduction to the variables section to make this clearer: “Information at baseline on gender, age, socioeconomic position, psychological demands and job control was collected via computer-assisted personal interview. At the end of the baseline and follow-up interviews, participants answered items on depressive symptoms privately in a separate paper questionnaire, which they gave back to the inter-viewer in a closed envelope to ensure confidentiality. Regarding the scales described below, quartiles, Cronbach’s Alphas and interitem correlations are shown in Table 1.”
- (line 100) As in my comment above, there are scores described for ‘subscales’ – what scale does this refer to? What would be considered a ‘high’ score? Can more information be provided please.
AUTHORS’ RESPONSE (page 4, lines 165-169): We have now clarified transparently what scales are main scales and subscales already in the first paragraph of the independent variables subsection. See also response to point 6.
- Line 107 – was the entire data collection based on computer assisted interview? Us tus tge sane as the SMGA?
AUTHORS’ RESPONSE: In the introductory paragraph of the Variables subsection, we indicate what modes of data collection were used. See our answer to point 8 above.
- Table 1: This is a little confusing to understand as half only have N and %, whereas the other half have Cronbach’s etc. Can this be split into two tables? More clarity is needed. Also for demographic information (e.g., age) means and standard deviations would be helpful.
AUTHORS’ RESPONSE (Page 5): Thank you for this comment. We suggest to keep all information in one Table to enable overview over the data. In order to avoid confusion, we now state clearly in Table 1 what variables were treated as categorical and what variables were treated as scales. Additionally we give the information on mean age in the text (Page 3, line 120).
- What is the COPSOQ inventory? What does this acronym stand for?
AUTHORS’ RESPONSE (Page 4, linews 155-160): We now write in the first lines of the independent variables section: “Psychosocial working conditions were measured with items from the 1st version of the Copenhagen Psychosocial Questionnaire (COPSOQ 1) [19, 20]. The COPSOQ instrument enables measurement of various constructs, including the DC Model. A comparison of the original items from the Job Content Questionnaire (JCQ) developed to measure t scales from the DC model, and the corresponding items from the COPSOQ instrument used in the present paper can be seen in Table 2.”
- Table 2: this could use some more explanation. I don’t understand the difference between the JCQ and the COPSOQ and why their items are being compared (?)
AUTHORS’ RESPONSE: See answer to point 12.
- The analysis section could be explained more clearly.
AUTHORS’ RESPONSE (Page 8, lines 241-248): We have now added introductory lines immediately under the 2.4. Analyses heading:
“We did two types of analyses in order to test assumptions of the DC model:
We performed interaction analyses in order to test if the observed combined effect of high psychological demands and low job control on depressive symptoms was stronger than their expected sum (i.e. superadditivity).
We performed differential subscale effect analyses in order to see if effects on depres-sive symptoms from the subscales of psychological demands (i.e. work pace and amount of work) and of job control (i.e. decision authority and skill discretion) were the same.”
- Results
The following text appears to be in error: This section may be divided by 15. subheadings. It should provide a concise and precise 250 description of the experimental results, their interpretation, as well as the experimental 251 conclusions that can be drawn.
AUTHORS’ RESPONSE (Page 8, between lines 281 and 282): We deleted these lines from the manuscript.
- Table formatting appears off in some tables
AUTHORS’ RESPONSE (Page 5, 10, 14 and 15): Now the text of the column headings is highlighted with bold characters in Tables 1 and 3 and Appendix Tables 1-3.
- Discussion
I suggest rearranging the order of your discussion to have the ‘comparison with other studies’ section before your limitations section.
AUTHORS’ RESPONSE (Pages 11-12, lines 344-385, not that we did not highlight these two subsections with yellow): Now, we have rearranged the order of these two sections in the discussion section.
- What if the participants were not in the same job during both data collection periods?
AUTHORS RESPONSE (Page 12, lines 392-396). In the analyzed population, we deliberately kept participants changing jobs or even exiting work at follow-up. We did that as depressive symptoms have been associated to long term sickness absence and disability pensioning. So keeping participants changing jobs or exiting work at follow-up contributes to avoid healthy worker selection in our analyses. We added this as a strength of the study in the discussion subsection. We now write: “Fourth, we followed up participants in the study disregarding if the still worked or exited work. Depressive symptoms have been associated to exit from work [32, 33]. Keeping participants changing jobs or exiting work at follow-up contributes to avoid healthy worker selection in the analyses of the present paper.”
- Why is the study described as being underpowered? To me this seems like a large enough sample size. Have you done power analyses?
AUTHORS’ RESPONSE (Page 12, lines 403-404): We now add in the discussion of weaknesses of the study the following sentence: “A post hoc power analysis yields that the present study has a power of 0.53 to find a significant elevated risk of 1.5 [34].”
- There is not a great deal of discussion of the present findings and their implications. I suggest including this so the reader understands what to take from these findings and how they may be applied.
AUTHORS RESPOSE (Pages 12 and 13, lines 415-436): Thank you for this comment. In the subsection ‘5. Conclusions’ we have now written a new first paragraph on practical implications of the present study´s findings. We also reworded the next paragraph on implications for research so as to make our messages here clearer.
Reviewer 2 Report
The theme is interesting.
Some suggestions:
- Table 1 - when the median is used to describe a quantitative variable, it is common to add the interquartile interval Median [Q1; Q3].
- Line 103: if the authors use the median to describe the variables "psychological demands" and "job control" then, the authors should use the Spearman coefficient instead of Pearson coefficient. And the author should indicate the respective p-value.
- Section 2.3.1.: the authors should present the Cronbach's Alphas of the translated and validated scales for comparison.
- Lines 163 and 164: this notation is very confuse!
- The authors don't say which software was used.
- 95% CI should be indicated as 95% CI [a; b] to be easier to read.
- Line 328: what is the power of this analysis?
- Line 428: problems on formatation text
- where is Figure 1?
Author Response
July 10th, 2021
ANSWERS TO COMMENTS OF REVIEWER 2
Thank you for your valuable comments improving the quality of the manuscript. Below are our point by point answers to each of your comments (with references to relevant pages and line numbers in parentheses). In the revised version of the manuscript, changes in text are highlighted with yellow background.
- Table 1 - when the median is used to describe a quantitative variable, it is common to add the interquartile interval Median [Q1; Q3].
AUTHORS’ RESPONSE (Page 5): We have added two columns in Table 1 showing 1st and 3rd quartiles.
- Line 103: if the authors use the median to describe the variables "psychological demands" and "job control" then, the authors should use the Spearman coefficient instead of Pearson coefficient. And the author should indicate the respective p-value.
AUTHORS’ RESPONSE (Page 3, lines 124-126): We think that is gives more information to offer the associations of the full scales lying behind the subsequent dichotomization. So we prefer to stick to reporting the Pearson correlation.
- Section 2.3.1.: the authors should present the Cronbach's Alphas of the translated and validated scales for comparison.
AUTHORS’ RESPONSE (Page 4, lines 147-149; Page 6, lines 190-191, 198-199 and 204-205): We now report published Cronbach’s alphas for published German versions of similar scales (PHQ-9 and some COPSOQ-based scales).
- Lines 163 and 164: this notation is very confuse!
AUTHORS’ RESPONSE (Page 4, lines 171-172): We deleted the use of numbers like 1 ⅓, and we deleted a => sign. The sentence now reads: “For the main analyses of subscale effects, subscales (work pace, amount of work, decision authority and skill discretion) were also dichotomized at their medians (Median values can be seen in Table 1) [2] and - for a sensitivity analysis (see the analysis subsection below) - divided into three categories based on the value ranges 0 – <1.333 for ‘Low’, 1.333 – <2.666 for ‘Medium’ and 2.333 – 4 for ‘High’.”
- The authors don't say which software was used.
AUTHORS’ RESPONSE (Page 8, lines 79-80): In the end of the analyses subsection we now mention software, version and command used.
- 95% CI should be indicated as 95% CI [a; b] to be easier to read.
AUTHORS’ RESPONSE (Text: Page 1, line 32, Page 8 and 9, lines 285 and 287; Tables: Page 10, 14 and 15): Thank you for this comment. We now use ; to separate CI’s in text (abstract and results section) and tables (Tables 3, 4, A1, A2 and A3).
- Line 328: what is the power of this analysis?
AUTHORS’ RESPONSE (Page 12, lines 404-405): We now add in the discussion of weaknesses of the study the following sentence: “A post hoc power analysis yields that the present study has a power of 0.53 to find a significant elevated risk of 1.5 [34].”
- Line 428: problems on formatation text
AUTHORS’ RESPONSE (Page 14, line 474): We now use italics to highlight the words ‘value based’ in the heading of Table A2.
- where is Figure 1?
AUTHORS’ RESPONSE (Page 3, lines 113-116): We are sorry to have forgotten to add the figures into the text. We added Figure 1 in the population subsection of the method section.
